# Automatic identification of a stable QRST complex for non-invasive evaluation of human cardiac electrophysiology

**Gunilla Lundahl[1], Lennart Gransberg[1], Gabriel Bergqvist[1], Göran Bergström[1,2], Lennart Bergfeldt**[1,3]*

1 Department of Molecular and Clinical Medicine, Institute of Medicine, Sahlgrenska Academy, University of Gothenburg, Gothenburg, Sweden, 2 Department of Clinical Physiology, Region Västra Götaland, Sahlgrenska University Hospital, Gothenburg, Sweden, 3 Department of Cardiology, Region Västra Götaland, Sahlgrenska University Hospital, Gothenburg, Sweden

* lennart.bergfeldt@hjl.gu.se

**Data Availability Statement:** The data underlying the findings in our study are not freely and directly available in a public repository because the original approval by the regional ethics board and the

## Abstract

### Background

A vectorcardiography approach to electrocardiology contributes to the non-invasive assessment of electrical heterogeneity in the ventricles of the heart and to risk stratification for cardiac events including sudden cardiac death. The aim of this study was to develop an automatic method that identifies a representative QRST complex (QRSonset to Tend) from a Frank vectorcardiogram (VCG). This method should provide reliable measurements of morphological VCG parameters and signal when such measurements required manual scrutiny.

### Methods

Frank VCG was recorded in a population-based sample of 1094 participants (550 women) 50–65 years old as part of the Swedish CArdioPulmonary bioImage Study (SCAPIS) pilot. Standardized supine rest allowing heart rate stabilization and adaptation of ventricular repolarization preceded a recording period lasting ≥5 minutes. In the Frank VCG a recording segment during steady-state conditions and with good signal quality was selected based on QRST variability. In this segment a representative signal-averaged QRST complex from cardiac cycles during 10s was selected. Twenty-eight morphological parameters were calculated including both conventional conduction intervals and VCG-derived parameters. The reliability and reproducibility of these parameters were evaluated when using completely automatic and automatic but manually edited annotation points.

### Results

In 1080 participants (98.7%) our automatic method reliably selected a representative QRST complex where its instability measure effectively identified signal variability due to both external disturbances ("noise") and physiologic and pathophysiologic variability, such as e.g. sinus arrhythmia and atrial fibrillation. There were significant sex-related differences in 24 of 28 VCG parameters. Some VCG parameters were insensitive to the instability value, while others were moderately sensitive.

informed consent from the subjects participating in the studies do not include such a direct, free access. If a reader wants access to the data underlying the present article for validation purposes, please contact Swedish National Data Service at snd@gu.se, referring to this study. The software used to process the electrocardiographic signals is developed on a platform owned by Ortivus AB, Danderyd, Sweden, by an agreement between the company and one of the authors (L. B.). Any inquiry regarding access to the software should be addressed to Per Karlsson, representing Ortivus, at per.karlsson@ortivus.com, and to the corresponding author.

**Funding:** This study was supported by the Swedish Heart and Lung Foundation (to LB # 20190652) and by grants from the Swedish state under the agreement between the Swedish government and the county councils, the ALF-agreement to LB (ALFGBG-722431). SCAPIS is supported by the Swedish Heart and Lung Foundation, the Knut and Alice Wallenberg Foundation, the Swedish Research Council and VINNOVA. The SCAPIS pilot study also received funding from the Sahlgrenska Academy at Gothenburg University and strategic grants from ALF/LUA in Western Sweden. The sponsors did not have any role in the study design, data collection and analysis, decision to publish, or preparation of the manuscript.

**Competing interests:** The authors have declared that no competing interests exist.

## Conclusion

We developed an automatic process for identification of a signal-averaged QRST complex suitable for morphologic measurements which worked reliably in 99% of participants. This process is applicable for all non-invasive analyses of cardiac electrophysiology including risk stratification for cardiac death based on such measurements.

## Introduction

Cardiac arrest and sudden cardiac death (SCD) is a major health problem and mainly due to an electrical disturbance although other etiologies exist [1]. Furthermore, electrical abnormalities such as wide QRS-T angles have in various cohorts shown prognostic value regarding cardiac events including SCD even beyond conventional demographic and clinical variables and are preferentially assessed from the Frank vectorcardiogram (VCG) [2–12].

The recording of the VCG differs mainly from standard electrocardiography (ECG) by placing one electrode on the back and one on the neck [11]. The VCG related information is based on three orthogonal leads (X, Y and Z) from which the P, QRS and T vector loops can be defined as well as one global QRST complex from $QRST_x$, $QRST_y$ and $QRST_z$; S1 Fig. Unique for VCG are: 1) measures of amplitudes and directions of the QRS complex and T wave, 2) measures of global heterogeneities (from here dispersion) and separately for depolarization (QRS) and repolarization (T), 3) the ST vector magnitude reflecting ischemia, 4) measures of QRS- and T-loop complexity, and 5) the most accurate definition of the spatial QRS-T angles [13]. Despite strong scientific evidence of its prognostic value, as reviewed in [5–7], assessment of the QRS-T angles and other VCG-based measures has not yet become clinically established. There are multi-factorial reasons which include lack of standardization of recording and analysis methodology, lack of easily accessible interpretations of data and user friendly presentation of their implications. In this study we focused on obtaining accurate VCG-based parameters and not on the diagnostic performance or specific clinical applications of VCG which have been dealt with before [5–7, 13]. An automatic and reliable procedure for obtaining VCG derived parameters is one requirement towards clinical application of VCG and was the rationale behind this study.

In preparation for a Swedish population based study on cardiovascular and pulmonary risk factors, the **S**wedish **CA**rdio-**P**ulmonary bio**I**mage **S**tudy (SCAPIS, a pilot SCAPIS study with 1111 participants was initiated where Frank VCG was part of the day-2 protocol (n = 1095) [14]. These recordings provided a suitable data base for developing an automatic method. The aim of the study was therefore to develop a robust operator independent method applicable and appropriate for use in the upcoming main SCAPIS as well as in other studies and eventually for easy application in a clinical setting. This aim was achieved by a process which is applicable for all non-invasive analyses of cardiac electrophysiology whether based on Frank VCG or not. A step towards clinical implementation of VCG derived parameters for prognostic and other purposes was taken.

## Materials and methods

### Study participants

The participants were enrolled on a population basis among people 50–64 years living in the city of Gothenburg 2012 aiming at similar proportions of women and men as described in

detail elsewhere [14, 15]. The total target population consisted of 24,502 individuals. An invitation letter to participate in SCAPIS was sent to 2243 randomly selected individuals from 6 residential areas, which were selected to represent opposite extremes of socioeconomic status. The overall participation rate was 50% (1111 of 2243) but varied between the areas [15]. VCG recording was performed in all 1095 day-2 SCAPIS pilot participants by study personnel (staff nurses). One recording failed due to unobserved electrode dislodgement. The remaining 1094 constitute the study group.

Fifty participants (27 women) volunteered a second recording which allowed reproducibility assessment in this subgroup. The second recording was performed at a time-point of the participants' own choice because of the very tight 2-day schedule they already had agreed to undergo. Their mean age (SD) was 57 (4) years. In 22 of them, recordings were performed with ≥1month's interval while in 28 the second recording was performed the same day as the first but after removal of the electrode patches and with another operator performing the procedure after a time interval.

The research project was approved by the Regional Ethics Committee in Gothenburg, Sweden, on December 8, 2016, #1009–16. All participants provided written consent to all data collection.

## VCG recordings and on-line analysis

The methodology followed the basic principles applied in our previous studies [16–18]. The protocol stipulated 5 minutes of supine rest with closed eyes and no conversation to allow heart rate stabilization and heart rate adaptation of ventricular repolarization before the VCG recording period of ≥5 min (preferably 8–10 min). The CoroNet II system (Ortivus AB, Danderyd, Sweden) was used. Five electrodes were positioned around the chest (one in the back) aiming at the level of the 5th rib's insertion on the sternum, one in the neck, and one each on the left and right hip [11]. The electrodes on the back and neck (for Z- and Y-leads) were placed beside the spinous processes to optimize comfort. Signals were sampled at 500Hz with an amplifier bandwidth of 0.03 to 170Hz and the orthogonal X-, Y- and Z-leads were calculated according to Frank [12]. From consecutive 10s-periods of cardiac cycles with dominant QRST-morphology and good signal quality, signal-averaged QRST complexes (saQRST) from all cardiac cycles fulfilling these criteria were calculated during the automatic on-line analysis of the recording to improve the signal-to-noise ratio.

## Automatic VCG post-analysis

The analysis software was developed from the CoroNet platform (Ortivus AB, Danderyd, Sweden) but using the tangent method for defining the end of the T wave which we have applied in beat-to-beat analyses of ventricular repolarization dynamics [16–18].

For manual correction of annotation points (calipers), a graphical interface was used showing the automatic annotation points marked in a presentation where the X-, Y- and Z-leads and the magnitude of the QRST complex are superimposed; S1 Fig. See also Glossary and definitions in the S1 Appendix.

In brief, the aim of the post-analysis procedure was to identify the most representative 10s-saQRST complex among those created during the on-line VCG recording. The key property of this complex would be its similarity to the surrounding complexes indicating stable conditions. The first step was therefore to identify stable parts of the recording defined as the presence of 7 consecutive 10s-saQRST complexes (70s of the recording). In the next step, qualified segments were selected when the middle 5 consecutive saQRST complexes out of the 7 each contained ≥3 cardiac cycles with dominant QRST morphology, representing 50s of the recording. An

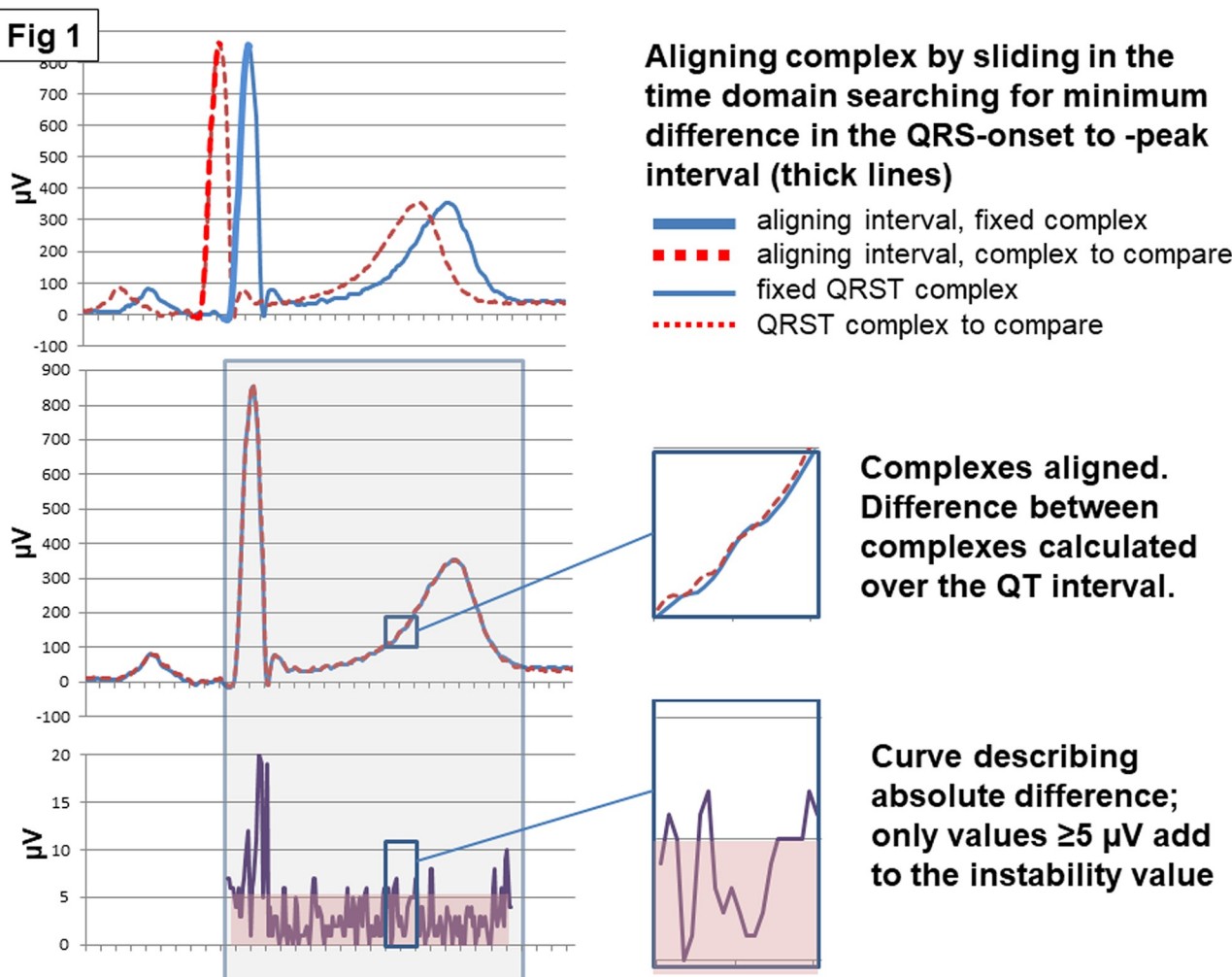

**Fig 1. Alignment process and difference calculation.** The upper panel shows the sliding and superimposition of the red QRST complex that was to be compared with the fixed blue QRST complex. The middle panel shows the alignment of the 2 QRST complexes and their difference in one lead in a short segment of the QT interval. The lower panel shows the calculation of the signal difference in the specific segment displayed in the middle panel. This procedure was used for 1) characterizing the stability of each 50s-segment of the recording (consisting of 5 consecutive 10s-signal-averaged complexes; saQRST complexes); 2) selecting the segment with least variability; 3) from the selected segment choose the 10s-saQRST complex which was most similar to the average complex of the selected 50s-segment. Comparisons were made at each 2ms-step (time resolution at sampling rate 500Hz) and the absolute amplitude difference over the QT interval was defined (entire QRST complex). The amplitude resolution was 2.5 µV and for each comparison all differences ≥5µV were used to calculate an instability value (no unit because small differences are not included).

instability value was calculated for each qualified segment based on the variability between its 5 10s-saQRST complexes. Finally, a representative 10s-saQRST complex was searched for among the 5 in the qualified segment with the lowest instability value. Representative in this context means the 10s-saQRST complex out of the 5 possible which was most similar to the average of the 5 complexes. The representative 10s-saQRST complex selected by this procedure was used for the calculation of all morphologic parameters. The process ending in the selection of a representative 10s-saQRST complex comprised 3 main steps utilizing 2 basic procedures. These procedures illustrated in Fig 1 were: 1) Alignment In order to compare QRST complexes or to create a saQRST complex, they were aligned by sliding and superimposing them in the time dimension, searching for the minimum difference in the interval between the QRS onset

and the QRS peak. By using the method of difference calculation contributions over the entire QT interval were given equal importance regardless of at what signal level they occurred, i.e. a difference of 10μV close to onset and a difference of 10μV close to the peak will have the same impact. 2) <u>Difference calculation</u> The difference between 2 complexes was defined as the mean difference of the sample values (one per 2ms) of the entire QT interval (QRSonset to Tend) after alignment as in 1). We focused on major deviations; any difference in an X-, Y- or Z-lead equal to just one shift in digitalization (2.5μV amplitude resolution) potentially only related to very little noise was therefore suppressed. Thus, all difference values less than 5μV in a lead were set to 0. This suppression was done before calculation of the sample difference value as $[dx^2+dy^2+dz^2]^{\frac{1}{2}}$ where dx, dy and dz are the differences in the X-, Y- and Z-leads. The difference calculation resulted in an instability value. The details of the 3-step process are described in the next paragraphs.

**1. Identifying qualified segments of the VCG recording.** The first step was to identify stable parts of the recording defined as the presence of 7 consecutive 10s-saQRST complexes (70s of the recording) with the dominant morphology and of good quality. The middle 5 of the 10s-saQRST complexes constitute a so called qualified segment. Although, the protocol stipulated 5 minutes of supine rest followed by ≥5 min of VCG recording, the first 2 minutes of the recording were avoided to ascertain optimal recording conditions. In addition, the final minute was excluded because it might contain disturbances if the participant became aware of the approaching end of the recording. When the recording period was <6 minutes, the discarded interval was gradually decreased to exclude half the recording period, 2/6 in the beginning and 1/6 of the recording at the end. When the available recording was <3 minutes or if no qualified segment was found in the selected part of the recording, any part of the recording including 5 consecutive 10s-saQRST complexes was accepted for further assessment.

**2. Selection of the qualified segment where complexes were most similar; variability calculation.** The variability of the signal was defined as the difference between the 5 consecutive 10s-saQRST complexes in a qualified segment of the recording. This difference was assessed by keeping one of the complexes fixed and the other 4 individually aligned. The difference between the fixed and aligned complexes was calculated and averaged as described in Fig 1. The procedure was repeated with each of the 5 saQRST complexes kept fixed and the lowest total difference of the 5 became the instability value assigned to the qualified segment. Every qualified segment was evaluated according to this procedure as illustrated by the flowchart in Fig 2. The qualified segment with the lowest instability value was selected as input for the final step and referred to as the selected segment.

**3. Selecting the representative 10s-saQRST complex for the entire recording.** The selected segment saQRST complex was calculated as the average of the 5 10s-saQRST complexes with the same one kept fixed (and the other 4 aligned) as gave the lowest instability value in step 2. All 5 10s-saQRST complexes in this segment were then aligned to the segment saQRST complex (i.e. to their average). The individual 10s-saQRST complex that had the smallest difference–was most similar—to the segment saQRST complex was selected as the representative 10s-saQRST complex for the entire recording and then used for the fully automatic measurement of 28 VCG derived parameters.

## Comparison of the representative algorithm selected vs. an arbitrary saQRST complex

The rationale behind this comparison was to test if a 5min pre-recording period of supine rest in itself resulted in subsequent recording segments of comparable stability as those identified through a rather extensive algorithm described in the previous sections. The 4th 10s-saQRST

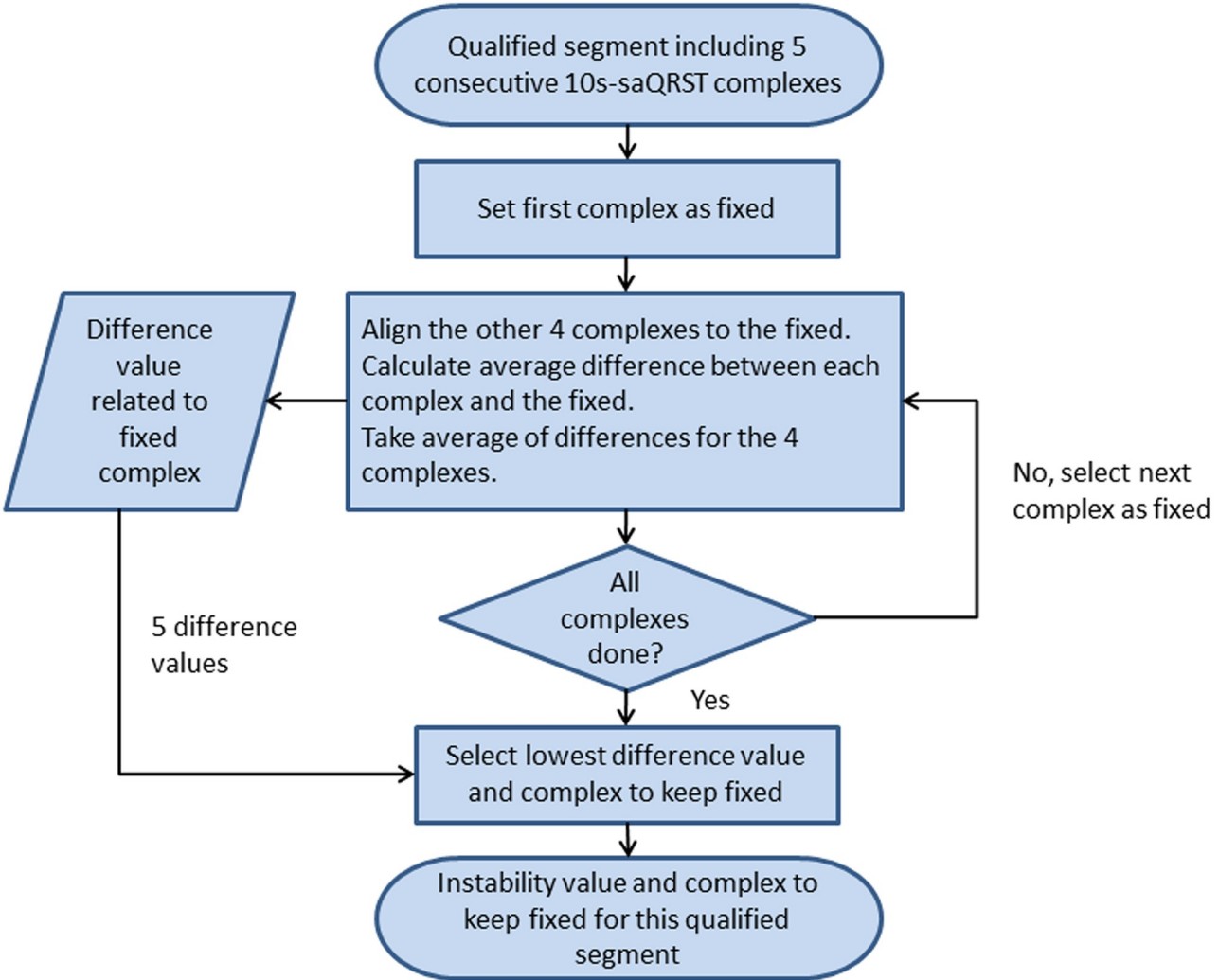

**Fig 2. Flow-chart showing the series of instability calculations.** The flow-chart describes the process for arriving at an instability value for a 50s qualified segment consisting of 5 consecutive 10s-signal-averaged-complexes (saQRST complexes) of the recording applying the methods described in Fig 1. This process was repeated for all qualified 50s segments of an individual recording and the segment with least variability was referred to as the selected segment from which the most representative 10s-saQRST complex was chosen for the calculation of 28 vectorcardiographic parameters.

complex from the beginning of each recording was arbitrarily chosen as comparator; i.e. representing an early part of the recording. This 10s-saQRST complex was analyzed in the same fully automatic way as the representative 10s-saQRST complex. We compared the actual parameter values as well as their reproducibility calculated as the coefficients of variation. Altogether 100 recordings (paired observations from the 50 participants studied twice) were eligible, but both recordings from one participant were excluded due to data not fulfilling the requisites of the automatic analysis.

### Assessment of fully automatic vs. manually edited automatic annotation points

In the same algorithm selected representative 10s-saQRST complex, we compared VCG parameters from automatically set annotation points vs. manually edited annotation points. In

order to avoid inter-observer variability one of the authors (L.B.) did the manual editing as described in the legend to S1 Fig. The same procedure as in the preceding paragraph was used with comparison of parameter values and the coefficients of variation.

## Quality validation, reasons for high instability values, and effect on VCG parameters

The instability value was both a tool in the selection process of the representative 10s-saQRST complex and a measure of its reliability or quality. The function for such a quality assessment was primarily to detect and signal technically unsatisfactory recordings that warranted manual editing by the investigator. We tested that this goal was achieved by 3 procedures. Two were manual and performed by one of the authors (G.L.) and the third was automatic. A) When there was a high instability value, the cause was categorized as due to: 1) external disturbances coming from technical problems or body movements, and 2) internal variations coming from breathing or variation in RR-intervals. Starting with the recordings with the highest instability values, a batch of 20–25 recordings were scrutinized. In a stepwise fashion another batch of recordings with successively lower instability values were scrutinized until a level was reached where the instability value had a low likelihood of being caused by external disturbances. B) The recording segment with the representative 10s-saQRST complex was compared with the rest of the recording to verify if it was from the most stable part in terms of disturbances, heart rate, and morphological parameters, and as a consequence suitable for morphologic analysis. C) Using the 5 10s-saQRST complexes in the selected segment, the relation between their instability value and their range (maximum–minimum) in the VCG parameter values (e.g. the QT interval) within this segment was tested by regression analysis.

## Statistical methods

Median and quartiles (Q1 and Q3) were used for descriptive purposes because most clinical variables and VCG parameters had non-Gaussian distributions according to the Shapiro-Wilk test with p-values <0.05. Between-group comparisons were performed with the Mann-Whitney and the chi-square tests. Reproducibility was assessed by the coefficient of variation (%), which was computed as the intra-individual standard deviation (s) divided by the mean of all values for each parameter (here 98 values) multiplied by 100 for percent. The intra-individual standard deviation was calculated as $[(\Sigma\, d_1^2/2+\ldots+d_n^2/2)/n]^{1/2}$ where d is the difference between the 2 paired observations in each of the 49 participants studied twice [19]. The Wilcoxon matched pairs test was used to analyze the comparisons of coefficients of variation. The Spearman rank order correlation coefficient ($r_s$) was calculated in the correlation analyses. A p-value < 0.05 was considered significant.

## Results

### Study participants

Table 1 presents demographic and clinical characteristics. There were 550 women and 544 men and their median age (Q1-Q3) was 57.6 (54.8–61.7) years without sex difference. The most common disorder among the participants was hypertension present in 364 of whom 53 also had diabetes while 33 had diabetes alone. Various other cardiovascular risk factors were also common. Men were not only taller and heavier but also had higher BMI and blood pressure and more often atrial fibrillation and diabetes (and hence higher blood glucose). Men also had higher triglycerides, Apo B/Apo A1, hemoglobin, ALT and creatinine. In contrast, women

**Table 1. Demographic and clinical characteristics of the population sample.**

| | All participants | Women | Men | |
|---|---|---|---|---|
| | n = 1094 | n = 550 | n = 544 | |
| Variable/parameter | Median (Q1-Q3) | Median (Q1-Q3) | Median (Q1-Q3) | p-value |
| Age [yrs] | 57.6 (54.8–61.7) | 57.5 (53.7–61.4) | 57.7 (53.9–62.0) | NS |
| Weight [kg] | 80.0 (69.0–90.0) | 70.4 (63.7–80.1) | 87.0 (80.0–96.0) | <0.001 |
| Height [m] | 1.71 (1.64–1.79) | 1.65 (1.60–1.69) | 1.78 (1.73–1.83) | <0.001 |
| BMI [kg m$^{-2}$] | 26.6 (24.4–29.4) | 26.0 (23.4–29.4) | 27.2 (25.2–29.6) | <0.001 |
| SBP left arm [mmHg] | 123 (114–135) | 121 (111–131) | 125 (116–137) | <0.001 |
| SBP right arm [mmHg] | 121 (112–132) | 118 (107–129) | 123 (114–134) | <0.001 |
| DBP left arm [mmHg] | 75 (68–81) | 72 (66–78) | 77 (72–83) | <0.001 |
| DPB right arm [mmHg] | 73 (68–80) | 71 (64–76) | 75 (70–82) | <0.001 |
| High blood pressure† [n (%)] | 220 (20) | 90 (16) | 130 (24) | <0.01 |
| **Disease history** | **n (%)** | **n (%)** | **n (%)** | **p-value** |
| Myocardial infarction | 12 (1.1) | 3 (0.5) | 9 (1.7) | NS |
| Coronary revascularization | 19 (1.7) | 5 (0.9) | 14 (2.6) | NS |
| Heart failure | 10 (0.9) | 2 (0.4) | 8 (1.5) | NS |
| Valve disease | 3 (0.3) | 1 (0.2) | 2 (0.4) | NS |
| Stroke | 11 (1.0) | 5 (0.9) | 6 (1.1) | NS |
| Atrial fibrillation‡ | 30 (2.7) | 9 (1.6) | 21 (3.9) | <0.01 |
| Hypertension | 364 (33.3) | 189 (34.4) | 175 (32.2) | NS |
| Diabetes | 86 (7.9) | 28 (5.1) | 58 (10.7) | <0.001 |
| Cancer | 80 (7.3) | 56 (10.2) | 24 (4.4) | <0.001 |
| Rheumatic disease | 74 (6.8) | 47 (8.5) | 27 (5.0) | <0.05 |
| Prescribed medication | 489 (44.7) | 262 (47.6) | 227 (41.7) | NS |
| **Smoking habits** | **n (%)** | **n (%)** | **n (%)** | **p-value** |
| Never smoked | 472 (43.1) | 245 (44.5) | 227 (41.7) | NS |
| Active smoker | 161 (14.7) | 81 (14.7) | 80 (14.7) | NS |
| Occasional smoker | 36 (3.3) | 23 (4.2) | 13 (2.4) | NS |
| Ex-smoker | 421 (38.5) | 198 (36.0) | 223 (41.0) | NS |
| **Blood analyses** | **Median (Q1-Q3)** | **Median (Q1-Q3)** | **Median (Q1-Q3)** | **p-value** |
| Cholesterol (total) [mmol L$^{-1}$] | 5.7 (5.0–6.5) | 5.9 (5.2–6.6) | 5.6 (4.8–6.3) | <0.001 |
| LDL [mmol L$^{-1}$] | 3.8 (3.1–4.4) | 3.8 (3.1–4.4) | 3.8 (3.1–4.4) | NS |
| HDL [mmol L$^{-1}$] | 1.6 (1.3–2.0) | 1.8 (1.5–2.2) | 1.4 (1.2–1.7) | <0.001 |
| Triglycerides [mmol L$^{-1}$] | 1.1 (0.8–1.6) | 1.0 (0.7–1.4) | 1.2 (0.9–1.8) | <0.001 |
| Apo B/Apo A1 ratio [unitless] | 0.66 (0.54–0.81) | 0.62 (0.51–0.74) | 0.71 (0.57–0.87) | <0.001 |
| Glucose [mmol L$^{-1}$] | 5.6 (5.2–6.1) | 5.5 (5.1–5.8) | 5.8 (5.5–6.3) | <0.001 |
| HbA1c [mmol L$^{-1}$] | 35 (33–38) | 35 (33–38) | 35 (33–38) | NS |
| Hemoglobin [g L$^{-1}$] | 140 (132–149) | 134 (127–139) | 148 (141–153) | <0.001 |
| ALT [μkat L$^{-1}$] | 0.43 (0.34–0.57) | 0.39 (0.31–0.50) | 0.48 (0.38–0.63) | <0.001 |
| hsCRP [mg L$^{-1}$] | 1.3 (0.6–2.8) | 1.4 (0.6–3.0) | 1.3 (0.7–2.6) | NS |
| Creatinine [μmol L$^{-1}$] | 78 (69–88) | 70 (63–76) | 86 (79–94) | <0.001 |

Characteristics of the population sample with available recordings and comparisons between women and men (Mann-Whitney test and $X^2$ with Yates correction).

Median (Q1-Q3). (<1% data missing for each item).

† systolic blood pressure ≥140 and/or diastolic blood pressure ≥90 mmHg

‡ including 4 newly diagnosed cases among 9 with this arrhythmia during the VCG recording

more often had cancer and rheumatic disease and had higher total cholesterol due to higher HDL (the beneficial lipoprotein).

The 12-lead ECG estimated from the Frank VCG was normal in 62% of the participants according to the evaluation performed by one of the authors (L.B.). Unspecific ST-T changes were the most common deviations from normality (9.9%), followed by premature ventricular extra-beats, fascicular blocks, and early repolarization which each of them was observed in 5–6% of participants. Sinus bradycardia, premature atrial extra-beats, first degree atrioventricular block, bundle branch block, and T-wave inversions was each of them observed in 1–4%, while pathological Q-waves, atrial tachycardia or atrial fibrillation and prolonged intraventricular conduction (QRS≥120ms) without typical bundle branch block pattern each was observed in <1%.

## Data acquisition and selection of representative 10s-saQRST complex

The protocol stipulated ≥5 minutes of recording time which was achieved in all but 10 participants (1084, 99.1%). Three recordings did not provide enough data for the automatic post-analysis and were therefore excluded. The median (Q1-Q3) duration of the remaining 1091 recordings was 9.2 (8.2–9.7) min with distribution according to S2 Fig. In 1059 recordings (97.1%) the first 2 and the last minute of the recording were excluded as planned, while in 32 recordings parts of these periods were included in the search for stable segments. In 3 cases, the entire recording was searched for 5 consecutive 10s-saQRST complexes.

In 11 recordings atrial activity was superimposed on the QRST interval, 9 due to atrial fibrillation and 2 due to competing sinus and junctional rhythms. In 5 out of 9 recordings with atrial fibrillation the instability value was >12 (cut-off limit chosen for reasons discussed below), and we decided to exclude all recordings with this arrhythmia as well as both recordings with competing sinus and junctional rhythms which both had instability values >12.

The automatic post-analysis included the remaining 1080 subjects with sinus rhythm (98.7%; 547 women, 533 men). VCG parameters from these 1080 participants were calculated from automatic annotation points on the representative 10s-saQRST complex (Table 2). Sex-related differences were the rule and observed for 24 (86%) out of 28 VCG parameters. Only QRSamplitude, QRSarea, QRSazimuth and QRSarea azimuth did not differ significantly between women and men. Most differences were, however, <5%, including longer QT and QTc duration in women as expected. The QRS duration was 7% larger in men. The direction of the QRS- and QRSarea-vector and T- and Tarea-vector was more cranial and the T-vector also directed more forward in the transversal plane in men than in women [measured as elevation from down-ward and up and as azimuth in the transversal plane from left towards right in a frontal (0 to 180˚) or dorsal (0 to -180˚) direction]. The dispersion parameters Tamplitude, Tarea and Ventricular gradient were larger (27, 36 and 14%) and the Peak and Mean QRS-T angles were wider in men (48 and 30%) (S1 Dataset). These VCG parameters thus pointed towards higher risk for cardiac events in men. The same pattern was observed when comparisons were made among 319 apparently healthy participants (151 women) without any acute or chronic disease, chronic medication, or pathologic blood tests; S1 Table.

## Comparison of representative algorithm selected vs. an arbitrary saQRST complex

The data from this comparison are shown in Table 3. Although these saQRST complexes were from different parts of the recording, the heart rate and the number of beats show that in general all cardiac cycles were of dominant morphology with good signal quality during the 10s-

**Table 2. Vectorcardiographic based parameters in 1080 women and men.**

| | All participants | Women | Men | p-value |
|---|---|---|---|---|
| | n = 1080 | n = 547 | n = 533 | |
| | Median (Q1-Q3) | Median (Q1-Q3) | Median (Q1-Q3) | |
| Heart Rate [bpm] | 68 (61–75) | 69 (63–76) | 66 (60–73) | <0.001 |
| PQ [ms] | 164 (150–182) | 162 (148–178) | 168 (154–184) | <0.001 |
| QRS [ms] | 94 (88–104) | 92 (86–102) | 98 (92–106) | <0.001 |
| QTpeak [ms] | 306 (290–324) | 312 (294–328) | 300 (284–316) | <0.001 |
| QT [ms] | 392 (372–412) | 396 (376–418) | 386 (368–408) | <0.001 |
| QTcB [ms] | 415 (398–435) | 424 (408–443) | 405 (391–425) | <0.001 |
| QTcF [ms] | 406 (392–424) | 413 (400–431) | 398 (387–415) | <0.001 |
| QTcFram [ms] | 406 (393–424) | 414 (401–432) | 399 (388–414) | <0.001 |
| QTcH [ms] | 406 (391–423) | 412 (399–430) | 398 (387–414) | <0.001 |
| Tpeak-end [ms] | 82 (76–92) | 82 (74–92) | 84 (76–94) | <0.01 |
| Tpeak-end/QT [unitless] | 0.21 (0.20–0.24) | 0.21 (0.19–0.23) | 0.22 (0.20–0.24) | <0.001 |
| QRSamplitude [mV] | 1.30 (1.05–1.59) | 1.28 (1.04–1.56) | 1.31 (1.06–1.64) | NS |
| QRSarea [µVs] | 29 (21–37) | 28 (21–36) | 29 (21–38) | NS |
| QRSelevation [°] | 57 (48–67) | 51 (44–61) | 63 (54–73) | <0.001 |
| QRSarea elevation [°] | 57 (45–71) | 51 (41–64) | 62 (51–78) | <0.001 |
| QRSazimuth [°] | 4 (-8-14) | 3 (-8-13) | 5 (-8-15) | NS |
| QRSarea azimuth [°] | -12 (-29-4) | -12 (-27-3) | -13 (-31-5) | NS |
| Tamplitude [mV] | 0.29 (0.22–0.39) | 0.26 (0.19–0.35) | 0.33 (0.25–0.43) | <0.001 |
| Tarea [µVs] | 39 (28–50) | 33 (25–44) | 45 (35–56) | <0.001 |
| Televation [°] | 53 (44–61) | 47 (40–54) | 58 (52–65) | <0.001 |
| Tarea elevation [°] | 53 (45–61) | 47 (40–55) | 59 (52–65) | <0.001 |
| Tazimuth [°] | 32 (19–45) | 27 (13–39) | 37 (27–50) | <0.001 |
| Tarea azimuth [°] | 41 (30–54) | 39 (25–52) | 44 (34–56) | <0.001 |
| Peak QRS-T angle [°] | 26 (16–43) | 21 (13–35) | 31 (20–50) | <0.001 |
| Mean QRS-T angle [°] | 46 (28–68) | 40 (24–59) | 52 (34–75) | <0.001 |
| Ventricular gradient [µVs] | 59 (46–77) | 56 (45–72) | 64 (50–83) | <0.001 |
| Tavplan [µV] | 0.35 (0.26–0.47) | 0.32 (0.25–0.42) | 0.36 (0.28–0.50) | <0.001 |
| Teigenvalue [unitless] | 31 (13–78) | 43 (15–93) | 23 (11–57) | <0.001 |

Frank vectorcardiogram parameters from automatic analysis of the 10s-signal-averaged QRST complexes in the 1080 participants of the population sample with comparisons between women and men (Mann-Whitney test). Median (Q1-Q3) (<1% data missing for each item).

sampling periods. The coefficients of variation for the arbitrarily chosen 4th "early" 10s-saQRST complex were larger for 13 out of 18 parameters. The largest improvements with the algorithm were observed for the conventional conduction intervals where the difference in CV was up to 50%. Overall, there was, however, no statistically significant difference when applying the Wilcoxon test for matched pairs (p = 0.14).

## Assessment of fully automatic vs. manually edited automatic annotation points

The data from this comparison performed on the representative 10s-saQRST complex are shown in Table 4. The unedited automatic annotations of the 10s-saQRST complexes gave similar results as manual editing of the annotations (Wilcoxon test for matched pairs, p = 1.0).

**Table 3. Comparison of the time-dependent variability for the representative algorithm selected vs. the 4[th] 10s-saQRST complex of the recording.**

| Data set: | 10s-saQRST complex automatic annotation algorithm selected | | 10s-saQRST complex automatic annotation 4[th]"early" | | CV difference |
|---|---|---|---|---|---|
| Parameter | x (s) | CV | x (s) | CV | % |
| Beat count [n/sample] | 11(1) | 8.7 | 11(1) | 8.4 | -3.1 |
| Heart rate [bpm] | 66 (4) | 6.0 | 67 (4) | 6.1 | 2.0 |
| PQ [ms] | 167 (6) | 3.7 | 166 (7) | 4.1 | 11.7 |
| QRS [ms] | 95 (5) | 4.9 | 94 (5) | 5.6 | 14.7 |
| QT [ms] | 392 (12) | 3.0 | 392 (14) | 3.7 | 20.6 |
| QTpeak [ms] | 308 (7) | 2.3 | 308 (8) | 2.4 | 4.5 |
| QTcB [ms] | 410 (13) | 3.1 | 413 (19) | 4.7 | 50.0 |
| Tpeak-end [ms] | 83 (11) | 13 | 84 (13) | 15 | 14.9 |
| Tpeak-end/QT | 0.21 (0.02) | 9.2 | 0.21 (0.02) | 10.8 | 17.1 |
| QRSarea [µVs] | 31 (4) | 13 | 31 (4) | 12 | -4.9 |
| QRSamplitude [mV] | 1.35 (0.15) | 11 | 1.35 (0.14) | 10 | -4.9 |
| Tarea [µVs] | 45 (8) | 18 | 46 (9) | 19 | 9.6 |
| Tamplitude [mV] | 0.35 (0.07) | 20 | 0.36 (0.06) | 18 | 9.8 |
| Peak QRS-T angle [°] | 34 (15) | 45 | 34 (16) | 47 | 3.9 |
| Mean QRS-T angle [°] | 48 (7) | 14 | 49 (7) | 15 | 6.0 |
| Ventricular gradient [µVs] | 69 (9) | 13 | 70 (10) | 14 | 5.7 |
| Tavplan [µV] | 0.36 (0.10) | 29 | 0.37 (0.10) | 27 | -7.9 |
| Teigenvalue [unitless] | 130 (262) | 201 | 120 (284) | 237 | 17.8 |

The sample mean (x), intra-individual standard deviation (s) and coefficients of variation (CV in %) for selected VCG parameters based on repeated recordings from 49 participants studied on 2 occasions for the algorithm selected representative saQRST complex and a randomly selected early (4[th]) saQRST complex. The sample mean is based on 98 observations to represent the denominator in the CV calculations. CV difference (in %) between CV(4[th]) and CV(algorithm).

## Quality validation, reasons for high instability values, and effect on VCG parameters

This part of the study was performed on the 1080 recordings qualifying for VCG measurements. The median (Q1-Q3) instability value, based on the 5 consecutive 10s-saQRST complexes in the selected qualified segment, was 4.2 (3.2–5.6) and their distribution is shown in S3 Fig.

Table 5 shows A) the reasons for high instability values, B) if the selected 50s- qualified segment was from the most stable part of the entire recording, and C) if the 10s-saQRST complex that had been selected was representative and of good quality. According to manual scrutiny of 70 recordings with the highest instability values, the higher the instability value, the more likely the reason was external disturbances. Furthermore, in 29 of the 33 recordings where the main reason for a high instability value was external disturbances, these were mainly caused by problems from the neck electrode (affecting the Y-lead in the orthogonal system); S4 Fig panel a. A high instability value, however, also reflected internal sources of variability due e.g. to breathing or RR variability; S4 Fig panel b. Manual scrutiny also confirmed that the automatic process had identified the 50s-recording segment from the most stable part of the recording. Finally, the selected 10s-saQRST complex was representative and of good quality for calculating the VCG parameters when the instability value was ≤12. Five recordings with instability values exceeding 12 were less satisfactory.

The relation between the instability value and the range (maximum–minimum) of VCG parameter values among the 5 10s-saQRST complexes within the selected 50s-segment are exemplified in Fig 3 (panels a-c for QT interval, Mean QRS-T angle, and Ventricular gradient)

**Table 4. Assessment of fully automatic vs. manually edited annotation points.**

| Data set: | 10s-saQRST complex automatic annotation | | 10s-saQRST complex edited annotation | |
|---|---|---|---|---|
| **Parameter** | **x (s)** | **CV** | **x (s)** | **CV** |
| **Beat count [n/sample]** | 11 (1) | 8.7 | 11 (1) | 8.7 |
| **Heart rate [bpm]** | 66 (4) | 6.0 | 66 (4) | 6.0 |
| **PQ [ms]** | 167 (6) | 3.7 | 165 (6) | 3.4 |
| **QRS [ms]** | 95 (5) | 4.9 | 98 (5) | 5.6 |
| **QT [ms]** | 392 (12) | 3.0 | 392 (10) | 2.5 |
| **QTpeak [ms]** | 308 (7) | 2.3 | 310 (7) | 2.4 |
| **QTcB [ms]** | 410 (13) | 3.1 | 410 (12) | 2.9 |
| **Tpeak-end [ms]** | 83 (11) | 13 | 82 (9) | 11 |
| **Tpeak-end/QT** | 0.21 (0.02) | 9.2 | 0.20 (0.02) | 8.4 |
| **QRSarea [µVs]** | 31 (4) | 13 | 31 (4) | 13 |
| **QRSamplitude [mV]** | 1.35 (0.15) | 11 | 1.35 (0.15) | 11 |
| **Tarea [µVs]** | 45 (8) | 18 | 45 (8) | 17 |
| **Tamplitude [mV]** | 0.35 (0.07) | 20 | 0.35 (0.07) | 20 |
| **Peak QRS-T angle [˚]** | 34 (15) | 45 | 34 (16) | 46 |
| **Mean QRS-T angle [˚]** | 48 (7) | 14 | 48 (7) | 14 |
| **Ventricular gradient [µVs]** | 69 (9) | 13 | 69 (9) | 13 |
| **Tavplan [µV]** | 0.36 (0.10) | 29 | 0.35 (0.10) | 30 |
| **Teigenvalue [unitless]** | 130 (262) | 201 | 137 (312) | 228 |

The sample mean (x), intra-individual standard deviation (s) and coefficients of variation (CV in %) for selected VCG parameters based on repeated recordings from 49 participants studied on 2 occasions. The sample mean is based on 98 observations to represent the denominator in the CV calculations.

with additional examples in the S5 Fig (panels a-e for QTpeak, Tpeak-end, Tamplitude, Tarea, and Peak QRS-T angle). The ranges in the ventricular repolarization dispersion parameters Tamplitude, Tarea and Ventricular gradient showed a moderate correlation with the instability value ($r_s^2$-values: 0.30, 0.25 and 0.26; $p<0.001$ for all). There was also statistically significant albeit weak biological correlations between the range in QTpeak (but not the entire QT interval), as well as for the Peak and Mean QRS-T angles on one side and the instability value on the other ($r_s^2<0.05$). Fig 3 and S5 Fig also show that in some individuals there were large ranges within the selected segment without relation to the instability value. Fig 4 illustrates

**Table 5. Recordings with high instability values.**

| | | Reason for high instability value | | Selected qualified segment from the most stable part of the recording? | Selected complex representative and of good quality? |
|---|---|---|---|---|---|
| **Instability value** | **Number of recordings** | **External disturbances** | **Internal variations** | **Yes** | **Yes** |
| | | **n (%)** | **n (%)** | **n (%)** | **n (%)** |
| >12 | 22 | 20 (91) | 2 (9) | 22 (100) | 17 (77) |
| 9.5–12 | 25 | 8 (32) | 17 (68) | 25 (100) | 25 (100) |
| 8.8–9.5 | 23 | 5 (22) | 18 (78) | 23 (100) | 23 (100) |

Data from stepwise assessment of the 70 recordings with highest instability values in the selected qualified 50s-segments with the representative 10s-saQRST complex.

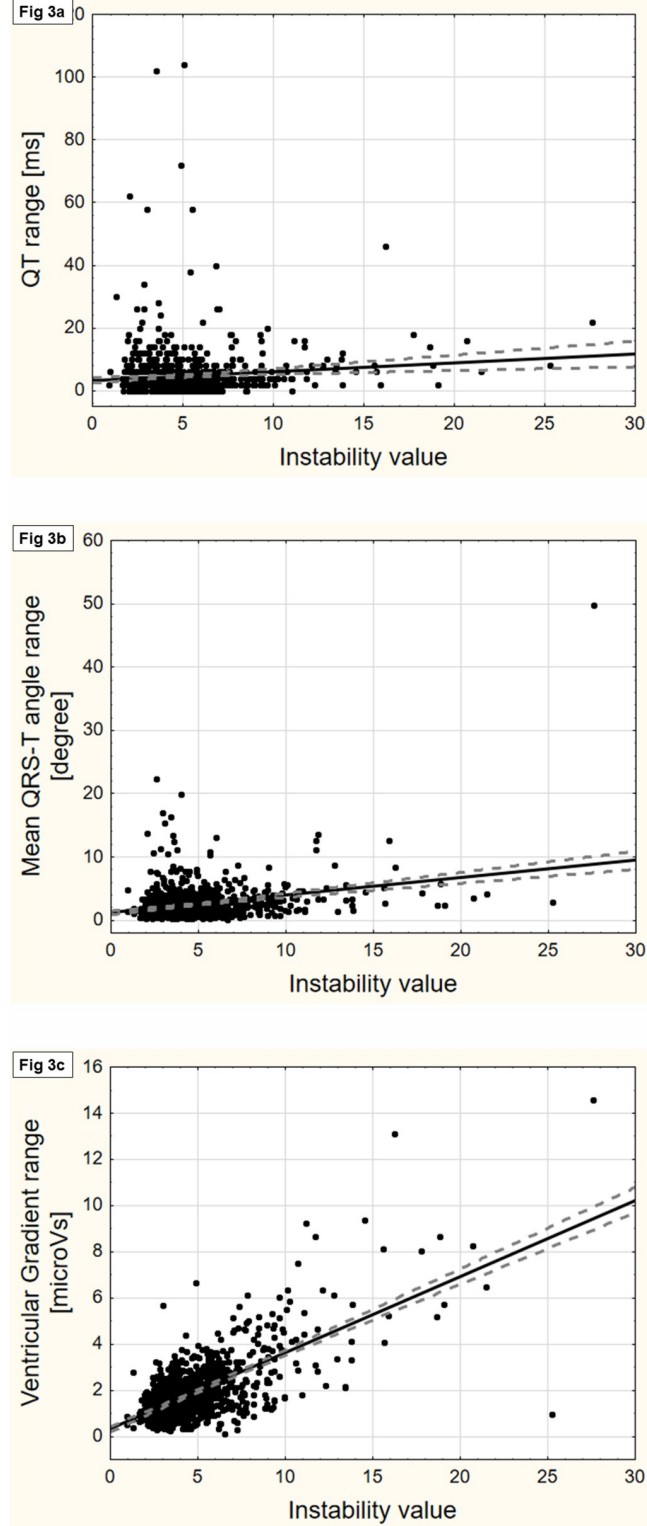

**Fig 3. The relation between the instability value and the range for 3 vectorcardiographic parameters.** Panels a-c show graphs of the relation between the ranges (maximum-minimum values) of vectorcardiographic parameters among the 5 consecutive 10s-saQRST complexes in the selected 50s-segment and its instability value (no unit); the QT interval (panel a; $r_s = 0.06$; NS; $r_s^2 < 0.01$), the Mean QRS-T angle (panel b; $r_s = 0.15$; $p < 0.001$; $r_s^2 = 0.02$), and the Ventricular gradient (panel c; $r_s = 0.51$; $p < 0.001$; $r_s^2 = 0.265$). $r_s$ is the Spearman rank order correlation coefficient. More examples are shown in S5 Fig.

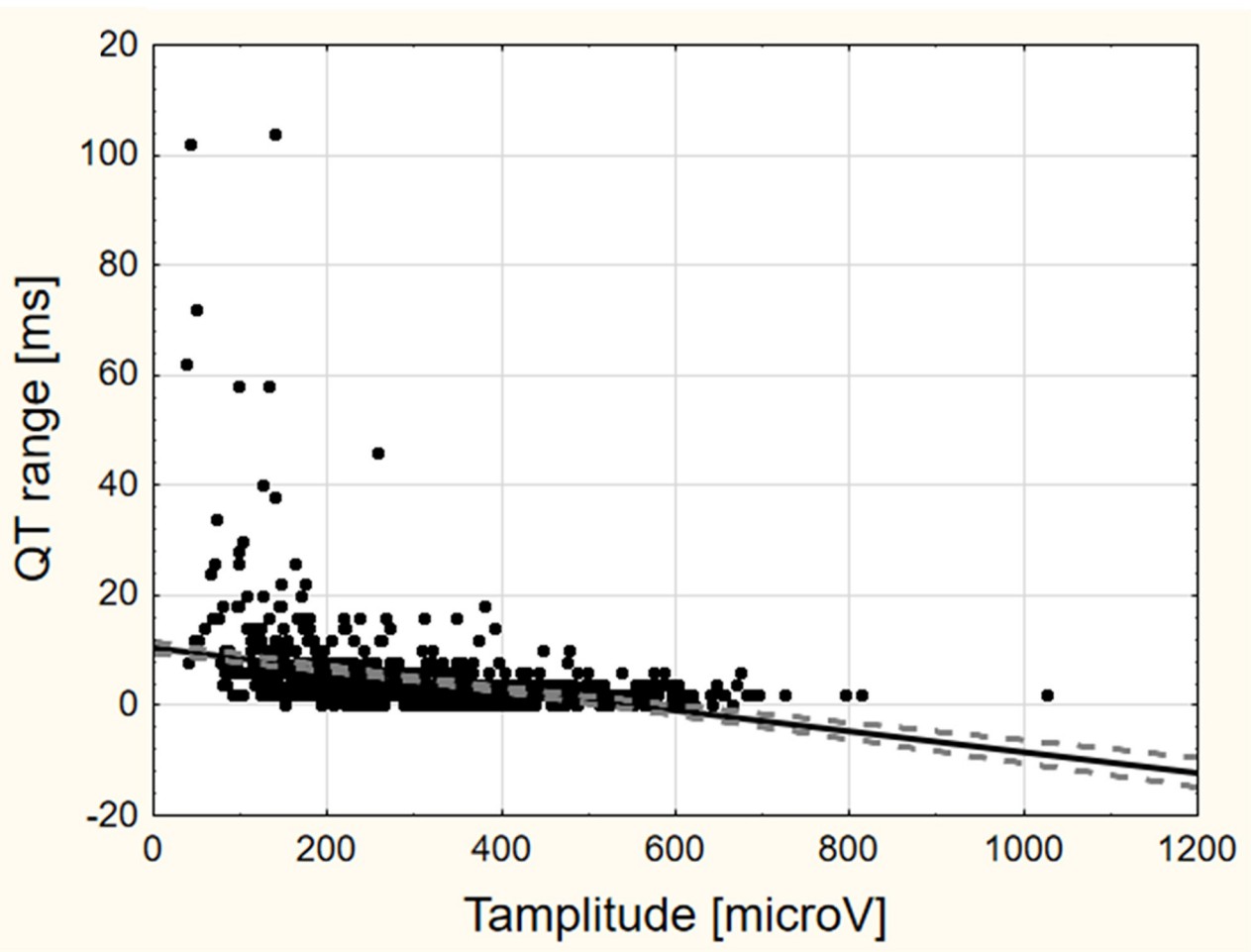

**Fig 4. The lower the T wave amplitude the greater the QT range.** This graph shows the inverse relation between the ranges in QT (maximum-minimum value) in the selected 50s-segment and the T-wave amplitude. The lower this amplitude the greater the differences in Tend and therefore also in the QT interval ($r_s$ = -0.48; $p < 0.001$; $r_s^2$ = 0.23). $r_s$ is the Spearman rank order correlation coefficient.

how the ranges in QT are inversely related to the Tamplitude; the lower the Tamplitude the larger the range in the QT interval, most obvious for Tamplitudes <200µV.

## Discussion

This study presents a method for automatic identification of a representative 10s-signal-averaged QRST complex from continuous Frank VCG recordings for the non-invasive evaluation of cardiac electrophysiology in humans. This method is, however, applicable for all non-invasive electrophysiological analyses whether based on Frank VCG or not. A quality measure (instability value) was implemented which identified the presence of external disturbances as well as physiological or pathophysiological variability, both potentially affecting VCG parameters and warranting manual scrutiny. The procedure proved feasible in 1080 (98.7%) of 1094 available recordings from a randomized population sample with equal proportions of women and men. The values of most VCG parameters differed significantly between women and men with the largest differences observed for the Peak and Mean QRS-T angles (48 and 30% larger in men), which are scientifically well-established risk-markers for cardiac death including SCD [2–11]. Men also had significantly larger heterogeneity (dispersion) of ventricular

repolarization measured as Tamplitude, Tarea and the Ventricular gradient, which might reflect their higher propensity for ventricular arrhythmias and sudden cardiac death.

Computerized electrocardiographic analysis is nothing new. An initiative for cooperation and standardization was published >30 years ago [the Common Standards for Quantitative Electrocardiography (CSE) project] [20]. Modern ECG equipment offer computerized calculations of PR, QRS and QT/QTc intervals as well as diagnostic interpretations. There is, however, an increasing interest in vectorcardiography-based analyses of electrocardiographic recordings for reasons summarized in the Introduction [13]. The literature on this topic describes results obtained with various customized computerized methods for VCG analyses mostly based on estimates from standard 12-lead ECG [5–7]. Focusing on the QRS-T angle, Schreurs et al. in 2010 reported the first validation of such estimates from standard 12-lead ECG when comparing 3 methods with Frank VCG as "gold standard" [4]. Even with the best method on the group level (referred to as the Kors matrix), there were in many cases considerable differences according to their Bland-Altman analysis [4; Fig 2]. Furthermore, and already within the CSE project, 19 computerized programs were compared, 10 based on 12-lead ECG and 9 on Frank VCG (XYZ); Willems and co-authors stated: "In general the measurement performance of XYZ programs was better than that of 12-lead programs." [20; p. 532]. The same authors commented on the importance of sufficient sampling size (recording duration) and on a high signal sampling rate using 500Hz in the CSE library as the standard. The present study was based on the Frank XYZ system, standardized supine rest during 5 minutes before the recording for ≥5min and using a sampling rate of 500Hz. We focused on obtaining high-quality data in both a technical and physiological sense but neither on the diagnostic performance in relation to specific diseases nor on comparisons between different vectorcardiography approaches or between VCG and 12-lead ECG; those issues are outside the scope of this study.

## Methodological aspects, limitations, and implications

The time-limiting step of the recording phase was defined by the hysteresis of the ventricular repolarization adaptation to a change in heart rate which is minimum 3 min [18, 21]. A 5 min resting period with closed eyes during silence therefore preceded a recording period ≥5min. This is different from the routines for recording clinical 12-lead ECGs, where the recording starts as soon as the electrodes have been attached and patient-related data has been entered into the recording system. In thorough QT testing for evaluation of the arrhythmogenic potential of pharmaceutical substances, however, a pre-recording period of 10 min is common (personal communication, Börje Darpö, MD PhD, Chief Scientific Officer, ERT®).

The goal of our procedure was an entirely unbiased process to identify stable 50s-periods of the entire recording for selecting at least one "qualified segment" free of noise and baseline drift. And within "the best" qualified segment (lowest instability value) select a representative 10s-saQRST complex for subsequent operator-independent calculation of VCG parameters. A fully automatic system also needs a built-in warning signal advising the user of the possibility of disturbances potentially affecting the precision/reliability of VCG parameters. Our quality measure–the instability value–serves this function by evaluating the variability between consecutive saQRST complexes. Such instability may, however, be due both to external sources (noise) and to physiological or abnormal pathophysiological variability. Increased heart rate (or RR) variability might be due to physiological sinus arrhythmia or to sinus node dysfunction [22]. Atrial fibrillation is another reason for increased and completely random RR variability. Atrial fibrillation may also affect the precision of the VCG parameters by the continuous atrial activity superimposed on the QRST complex and not cancelled out by

signal-averaging in the QRST complexes. Furthermore, increased ventricular repolarization variability is a salient feature of the long QT syndrome and other arrhythmia prone conditions [23–25]. Guided by the instability value, 70 recordings with the highest values were picked for manual scrutiny. The higher value, the more likely was external disturbances the source. Among the 48 recordings with values between 8.8 and 12, no selected 10s-saQRST complex had disturbances due to external sources. We therefore suggest that recordings with an instability value >12 should be manually inspected for signal quality and source of variability. Atrial fibrillation (9 recordings) and atrial flutter (although not observed in this study) as well as competing sinus and junctional rhythms (as in 2 participants in this study) are sources of signal instability. A high instability value might therefore signal the presence of such arrhythmias, which a closer inspection of the recording will confirm.

Assuming that not all VCG parameters would be equally sensitive to signal instability, we also performed an analysis regarding the relation between the instability value and the ranges of some VCG parameters reflecting ventricular repolarization duration and dispersion and the QRST-angles, as illustrated by Fig 3 and S5 Fig. The dispersion parameters were more sensitive to signal variability than the QT and QTpeak intervals as well as the QRS-T angles. This could be expected in view of the chosen procedure for calculating the instability value which was based on differences in sample values (Fig 1). Furthermore, the annotation point for the end of the T-wave, and consequently the value of the QT interval, was sensitive to the amplitude of the T-wave. This result corroborates previous observations by e.g. Vink et al. that a low and flat T-wave affects the measuring precision of the QT interval [26]. A low Tamplitude may therefore serve as another warning signal warranting manual scrutiny of the recording. When the Tamplitude exceeded 200μV there was rarely a problem in this study.

Our ultimate goal for the development of a computerized/automatic VCG analysis is to provide its user with reliable VCG parameters for clinical purposes such as risk prediction regarding cardiac death including sudden cardiac death [2–11]. A suitable risk marker should have as good reproducibility as possible, including all technical aspects and the individual time-dependent variability. Calculating the coefficient of variation is one alternative for such assessment and independent of units, which we have used in the electrophysiological context before [19]. A coefficient of variation <10% is usually considered very good or excellent. The presented method gave lower coefficients of variation compared to selecting an early part of the recording. Furthermore, the spatial Peak and Mean QRS-T angles are scientifically but not clinically established risk factors for cardiac death and sometimes grouped together [2–11]. In this study, the Mean QRS-T angle had much better reproducibility than the Peak QRS-T angle (coefficient of variation 14 vs. 45%), which favors the former for risk assessment. A recent study from our group, however, suggests that both angles together rather than one of them alone should be used for risk evaluation, which is one clinical context in which VCG can contribute valuable and accurate information [11, 13]. Another potential application would be in the prediction of the response to cardiac resynchronization therapy for heart failure and the timing of stimulation intervals [27, 28].

The QT interval has a scientifically and clinically established position as risk marker, especially in patients with the long QT syndrome [26]. The VCG based QRST complex allows the measurement of the QT interval unaffected by the T loop axis, which varies individually with regard to any lead on the ECG. It thus meets the requirement for the global QT interval which some scientist advocate and try to obtain by measuring the interval between the first QRS onset and the last Tend in any of the 12 leads of the standard ECG [29]. It has also been shown that the global QT interval calculated from Frank VCG differentiates between LQTS mutation carriers and age- and sex-matched controls better than QT intervals from either automatically or manually assessed standard 12-lead ECG [30]. We showed that the global QT and QTc

intervals had an excellent reproducibility at the 3%-level. The tangent method for defining the end of the T-wave, initially introduced as an alternative in special cases [31] gives a slightly shorter QT interval (10-15ms) than using the so called "threshold method" [26]. While some experts recommend both methods e.g. facilitated by a web-based QT calculator [26, 31], the tangent method has been favored by others [32]. The tangent method is more suitable than the threshold method for studying ventricular repolarization changes on a beat-to-beat level during rapid heart rate increase according to our own experience [16–18]. We therefore decided on the tangent method for the entirely automatic VCG analysis of saQRST complexes so that the same automatic method could be used for analyses of beat-to-beat and steady-state signal-averaged cardiac cycles.

The participants in this study represent a population based sample 50 to 64 years old at enrolment and up to 65 years at completion of the protocol. The general idea of the SCAPIS study is to obtain various risk markers for cardiovascular and pulmonary disease at an age where interventions supposedly are able to change the outcome in a favorable direction [14]. If new risk markers provide additional prognostic value on top of the already established and can be amended is an ongoing part of SCAPIS but outside the scope of this study. We cannot rule out that our method when applied to a sicker cohort would need manual scrutiny in more recordings than in the present. In 38% of these recordings there were, however, some abnormalities, and when using an instability value >12 and a low Tamplitude (< 200μV) as signals of possible imprecision of the VCG parameters, we anticipate this method to work in any clinical cohort.

For a century it has been known, and part of clinical practice, to take into account the sex related difference in the QT interval [33]. This study illustrates that most VCG-based parameters of cardiac electrophysiology show sex-related differences on the group level which should be taken into account in clinical studies. The Peak QRS-T angle was 48% and the Mean QRS-T angle 30% wider in men than in women and an age- and sex-related difference in these and other VCG parameters has been reported before [6, 8, 34]. Our entire study cohort includes participants with various cardiovascular and other diseases as well as chronic medication with various substances in almost half of them, which potentially might affect the VCG parameters, e.g. diabetes and hypertension [11]. The sex-related differences remained, however, when comparing data from the 151 women and 168 men without any known diseases or chronic medication in our cohort. In a previous Frank VCG based study on LQTS patients and age- and sex-matched controls, which were on average between 30 and 40 years of age, similar sex-related differences were observed [34].

Data from the present and the previous study may serve as a reference for apparently healthy men and women. Compared to a study from 1964, our data in healthy men were similar with regard to the comparable parameters QRSarea, Tarea and Ventricular gradient but the QRS-T angle was narrower in our men [35].

We have developed a method for recording and reliable analysis of Frank VCG which is ready for application in epidemiological studies such as the SCAPIS main study. Further development is, however, needed to achieve computerized on-line analysis with presentation of data and interpretation of their potential clinical implications before bedside use can be realized.

## Conclusion

A reliable automatic method to acquire Frank VCG parameters reflecting cardiac electrophysiology for risk stratification and other purposes was developed. The method includes a quality measure which informs the user of signal variability that warrants manual scrutiny. The

procedure proved feasible in 98.7% of 1094 participants in a population based cohort. Most VCG parameter values differed significantly between women and men and this difference should be taken into account in future studies.

## Supporting information

**S1 Appendix. Glossary and definitions.**
(DOCX)

**S1 Fig. User interface for manual editing of annotation points.** Graphical user interface for manual editing of annotation points, in this example with the green cursor at the QRSoffset, i.e. the J-point. Four leads of the QRST complex are shown, the X-, Y- and Z-leads and an averaged vector magnitude lead in white (Mag for magnitude) providing the "global" QRST complex.
(DOCX)

**S2 Fig. Recording duration.** Frequency distribution histogram of the recording duration in full minutes from 1091 participants. The median (Q1-Q3) was 9.2 (8.2–9.7) min. Data show non-Gaussian distribution (Shapiro-Wilk test). This graph shows that in almost all participants sufficiently long recording segments were available for analysis according to the algorithm.
(DOCX)

**S3 Fig. Instability values.** Frequency distribution histogram of instability values (no unit) rounded to nearest whole number for 1080 automatically selected 50s-segments (each consisting of 5 10s-saQRST complexes). Median (Q1-Q3) was 4.2 (3.2–5.6). Data show non-Gaussian distribution (Shapiro-Wilk test). Values > 12 suggest that the recording should be manually scrutinized; see text and Table 5 in the main manuscript for more details.
(DOCX)

**S4 Fig. Causes of high instability value.** Recordings with high instability values (>8.8) were scrutinized to find its cause and to categorize it as either external to the study subject (noise) or internal and of physiological or pathophysiological origin. Panel a: ECG from a section within the selected segment with disturbances on the Y-lead (originating from the neck electrode); this cause is defined as "external". Panel b: ECG from a section within the selected segment with varying RR-intervals due to physiological sinus arrhythmia in a study subject with relatively low heart rate; this cause is defined as "internal".
(DOCX)

**S5 Fig. The relation between the instability value and the range for 5 vectorcardiographic parameters.** Panels a-e show graphs of the relation between the ranges, (maximum—minimum values), of vectorcardiographic parameters within the selected 50s-segment and its instability value (no unit); QTpeak interval (panel a), Tpeak-end interval (panel b), Tamplitude (panel c), Tarea (panel d) and Peak QRS-T angle (panel e). These graphs show that in some individuals there were considerable variations in specified parameters despite low instability suggesting absence of external disturbances ("noise"). $r_s$ is the Spearman rank order correlation coefficient.
(DOCX)

**S1 Table. Vectorcardiographic parameters in apparently healthy women and men.** Vectorcardiographic parameters in the sub-group of 319 apparently healthy participants among the population sample of 1080 with comparisons between women and men (Mann-Whitney test).

Median (Q1-Q3) (<1% data missing for each item). Reference values for the age group 50–65 years.
(DOCX)

**S1 Dataset. Vectorcardiographic parameters, sex, and age.**
(PDF)

## Acknowledgments

We are very grateful to all the participants in this study and the staff at the SCAPIS test center in Gothenburg.

## Author Contributions

**Conceptualization:** Gunilla Lundahl, Lennart Gransberg, Lennart Bergfeldt.

**Data curation:** Gunilla Lundahl, Lennart Gransberg.

**Formal analysis:** Gunilla Lundahl, Lennart Gransberg, Lennart Bergfeldt.

**Funding acquisition:** Göran Bergström, Lennart Bergfeldt.

**Investigation:** Göran Bergström, Lennart Bergfeldt.

**Methodology:** Gunilla Lundahl, Lennart Gransberg, Lennart Bergfeldt.

**Project administration:** Göran Bergström, Lennart Bergfeldt.

**Resources:** Göran Bergström, Lennart Bergfeldt.

**Software:** Gunilla Lundahl, Lennart Gransberg, Gabriel Bergqvist.

**Supervision:** Göran Bergström, Lennart Bergfeldt.

**Validation:** Gunilla Lundahl, Lennart Gransberg, Lennart Bergfeldt.

**Visualization:** Gunilla Lundahl, Lennart Gransberg.

**Writing – original draft:** Gunilla Lundahl, Lennart Bergfeldt.

**Writing – review & editing:** Lennart Gransberg, Gabriel Bergqvist, Göran Bergström, Lennart Bergfeldt.

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
