## [Decision Letter · Decision Letter 0]

26 Jun 2020

PONE-D-20-11644

Automatic identification of a stable QRST complex for non-invasive evaluation of human cardiac electrophysiology

PLOS ONE

Dear Dr. Bergfeldt,

Thank you for submitting your manuscript to PLOS ONE. After careful consideration, we feel that it has merit but does not fully meet PLOS ONE’s publication criteria as it currently stands. Therefore, we invite you to submit a revised version of the manuscript that addresses the points raised during the review process.

Please address comments indicated by the Reviewers and shorten the length of the manuscript .

We look forward to receiving your revised manuscript.

Kind regards,

Elena G. Tolkacheva, PhD

Academic Editor

PLOS ONE

Journal Requirements:

Reviewers' comments:

Reviewer's Responses to Questions

**Comments to the Author**

1. Is the manuscript technically sound, and do the data support the conclusions?

Reviewer #1: Partly

Reviewer #2: Yes

2. Has the statistical analysis been performed appropriately and rigorously? 

Reviewer #1: I Don't Know

Reviewer #2: Yes

3. Have the authors made all data underlying the findings in their manuscript fully available?

Reviewer #1: No

Reviewer #2: Yes

4. Is the manuscript presented in an intelligible fashion and written in standard English?

Reviewer #1: No

Reviewer #2: Yes

5. Review Comments to the Author

Reviewer #1: In this manuscript, the authors describe a method for recording vectorcardiograms (VCG) in a population sample. The conclusion is that with a 5 minute recording period, 99% of subjects can have an adequate tracing produced.

The introduction attempts to cover all the relevant history of the vectorcardiogram. It could be shortened; the first paragraph is duplicative as the prognosis of VCG is repeated in the 2nd paragraph. The first 2 sentences of the 2nd paragraph could be removed. Instead, it might be nice for the reader to get a brief introduction to how/why the VCG is different from the standard ECG.

Methods: please elaborate on what is meant by “randomized fashion”. How were participants recruited? What was randomly assigned? Do the author mean to say that a random sample of people from the population were recruited? If so, how many were they drawn from, how was randomization performed, what were the power calculations to determine sample size, how many people elected to or declined to participate?

The distribution of normal/abnormal findings on VCG sounds similar to my clinical practice, are there other population samples for prevalence of abnormal ECG that can be referenced for comparison?

If the goal of this investigation is to demonstrate the usability of this technique, the discussion should be more focused on that aspect. Would recommend more attention be given to clinical application of this technique. For example, there are 6 pages of methods on the technique, how translatable are those methods to clinical practice? If the median time to acquire a VCG is 9 minutes (longer than the average face to face time for a patient and a physician in most office visits), how practical is it? Is the additional prognostic information worth the time/effort?

Reviewer #2: This study aimed to apply a novel method for standardizing the window of analysis for vectorcardiography in a large sample of patients in an epidemiological study in Sweden. Vectorcardiography is based on analysis of 12-lead ECG or Frank VCG recording leads and employs vector-based analyses (i.e., magnitude and angle) of recorded electrical signals. Though not readily employed clinically, the potential of VCG is convincingly great. This specific study in this space was unique, innovative, and largely technically appropriate. The authors demonstrated success of their algorithm in comparison to a randomly selected and manually annotated vectorcardiogram, success in delineating differences between men and women that they saw in their demographic analysis, consistency in serial visits with the same patient, and success in consistently producing a stable signal-averaged QRS complex and T wave for analysis with standard VCG tools. Additionally, the authors recognized the limitations of their study and drew mostly appropriate conclusions from their results. However, there is a major concern about the motivation and significance of this research. The authors did not fully elucidate the need for such an algorithm nor did they convincingly demonstrate how the application of their algorithm will aid in clinical care. These issues and some technical issues listed below should be considered before acceptance:

Major Comments

• Introduction

1. Greater discussion on the application of VCG in clinical scenarios, uniqueness compared to ECG (i.e., what complementary or supplementary information is given by the VCG), and prognostic value need to be established. These are vaguely described in Lines 87-100, but a clear description of VCG, its parameters, and its merits/disadvantages need to be included, especially given its nonstandard use. A diagram explaining VCG would be helpful, though not necessary as there are various other sources that the authors include for this. Additionally, the problem with manual selection of VCG segments must be delineated. Without these descriptions of VCG, the significance, novelty, and innovation of the authors’ study is lost.

• Methods

1. If it is possible to do, the variability between multiple observers (at least two) in manually edited annotation points is an important parameter to include to help demonstrate the success of your automatic algorithm.

2. What is the importance of the instability value and how do you actually calculate it? You discuss that the variability value is calculated from averaging the difference between the fixed and alignment waveforms, and that values of difference greater than 5 uV are used to calculate the instability value, but the actual calculation of instability is unclear. Additionally, the physiologic importance is not well established other than vaguely in Table 5 where “external disturbances” and “internal variations” are mentioned as contributing factors to instability. The process of manually defining something as “external” or “internal” is suspect without a more complete and rigorous analysis. Please describe how you delineated this.

• Results

1. The significance of a percentage difference in CV in Table 3 is unclear. Though a larger difference in CV would show that there is greater variability in the manual annotation on the fourth saQRST than in your automated algorithm, the significance of a 11.7% vs. 14.7 % difference is unclear. An interpretation or a separate statistic that indicates significant differences would be helpful. Additionally, it is assumed that there will be some inter-beat variability regardless of whether the automated or manual technique is used. Is part of the CV difference due to this inter-beat variability?

2. As discussed in Methods Comment 2, the reasoning behind classifying an instability value as due to external disturbances or internal sources of variability is essential. Is there a way to demonstrate what each of these would look like in a representative trace? Or are there criteria that your VCG reader used to define this?

• Discussion

1. One of the major questions that remains despite the success of your algorithm/study is the applicability of this technique. First, as you note in the paper, the typical ECG/VCG protocol does not include 5-10 minutes of recording so using this algorithm seems to potentially be impractical/infeasible. Second, because you chose not to look at how specifically the algorithm did in those with underlying heart disease (especially arrhythmias), the importance and utility of the algorithm is unclear. This is especially true as the VCG is expected to help diagnose transient arrhythmias like AF. As such, the success of the algorithm to identify a stable region of the VCG in these conditions is important, and I am not convinced that the author’s algorithm is capable of accomplishing this.

Minor Comments

• Abstract

1. QRST complex is not a standard complex according to typical ECG where there is a QRS complex and a T wave. Consider defining this complex in the abstract so the reader understands.

• Methods

1. Why was the second visit non-standardized? Would this have an effect on secondary results?

2. Where on the neck and back are electrodes placed? It would be helpful to include anatomical landmarks nearby like the vertebrae, vasculature, etc.

3. What is a signal-averaged QRST complex? Some specifics on this calculation would be helpful as it is hard to understand how 70 seconds could represent 7 10s-saQRST complexes and the order of operations that produces a signal averaged complex is unclear. Additionally, is this averaging the same or different than the process you describe for selecting the representative complex?

4. Can you explain why a sliding absolute difference between a portion of the QRS complex was used rather than a cross-correlation for the alignment steps?

5. Was there a reason that 5 uV was selected as the threshold to add to the instability value?

6. A quick question regarding the annotation. It is clear that the algorithm in this paper is focused on selection of a standard window for analysis of VCG parameters. However, the automatic/manual annotation of these parameters is not clear. Can you describe briefly how these parameters are selected and how sensitive the process is to noise? Perhaps using Supplemental Figure 1.

• Results

1. Table 1 is an excellent representation of demographics and shows clear differences in biological sex and various parameters in the study. Additionally, Table 2 excellently portrays the difference in various VCG parameters between men and women. A tertiary analysis of other demographics and VCG parameters seems warranted (especially since the data is available).

2. What were the values of the Shapiro Wilk test for normality and how did you decide to accept the null hypothesis (i.e., what value of the Shapiro Wilk test was used for this)? Qualitatively, there appears to be a relatively normal spread so more clarity would be appreciated.

3. Is there a statistic to effectively compare the CV, mean, or s.d. of the two different annotations in Table 4? If so, this information would be useful to the reader to understand the significance of this automated annotation.

6. PLOS authors have the option to publish the peer review history of their article (what does this mean?). If published, this will include your full peer review and any attached files.

Reviewer #1: No

Reviewer #2: No

---

## [Author Response · Author response to Decision Letter 0]

23 Jul 2020

RESPONSES TO REVIEWERS’ COMMENTS

PONE-D-20-11644

Automatic identification of a stable QRST complex for non-invasive evaluation of human cardiac electrophysiology

PLOS ONE

We very much appreciate the interest in our study and the reviewers’ suggestions to improve the manuscript. The numbers of the lines referred to in the responses are those in the manuscript with changes in track mode.

Reviewers' comments:

Reviewer's Responses to Questions

Comments to the Author

1. Is the manuscript technically sound, and do the data support the conclusions?

Reviewer #1: Partly

Reviewer #2: Yes

2. Has the statistical analysis been performed appropriately and rigorously? 

Reviewer #1: I Don't Know

Reviewer #2: Yes

3. Have the authors made all data underlying the findings in their manuscript fully available?

Reviewer #1: No

Reviewer #2: Yes

4. Is the manuscript presented in an intelligible fashion and written in standard English?

Reviewer #1: No

Reviewer #2: Yes

5. Review Comments to the Author

Reviewer #1: In this manuscript, the authors describe a method for recording vectorcardiograms (VCG) in a population sample. The conclusion is that with a 5 minute recording period, 99% of subjects can have an adequate tracing produced.

RESPONSE It appears that the present conclusion in the abstract is somewhat misleading and it has therefore been rephrased, lines 52-56. Actually, the main message is rather that we created an automatic process to identify a representative QRST complex, as stated in the title, based on the most stable 10s-segment from the 5 min recording. All QRST complexes in this segment (10 at a heart rate of 60 bpm etc.) were then averaged to improve the signal-to-noise ratio, a common procedure in signal analysis. From this signal-averaged QRST complex we measured automatically 28 VCG parameters. This procedure could, however, also be used for 12-lead electrocardiograms etc.

The introduction attempts to cover all the relevant history of the vectorcardiogram. It could be shortened; the first paragraph is duplicative as the prognosis of VCG is repeated in the 2nd paragraph. The first 2 sentences of the 2nd paragraph could be removed. Instead, it might be nice for the reader to get a brief introduction to how/why the VCG is different from the standard ECG.

RESPONSE The point is taken. The second paragraph of the Introduction has consequently been reconstructed and information on VCG vs. ECG differences is incorporated. There are also some changes in the third/final paragraph of the Introduction.

Methods: please elaborate on what is meant by “randomized fashion”. How were participants recruited? What was randomly assigned? Do the author mean to say that a random sample of people from the population were recruited? If so, how many were they drawn from, how was randomization performed, what were the power calculations to determine sample size, how many people elected to or declined to participate?

RESPONSE Yes, a random sample from the population was recruited. This process has now been elaborated upon in the text, lines129-132, and a reference (#15) which specifically deals with this process has been added.

The distribution of normal/abnormal findings on VCG sounds similar to my clinical practice, are there other population samples for prevalence of abnormal ECG that can be referenced for comparison?

RESPONSE We have searched for such a study but not found any with similar age- and sex-distribution. Turned the other way around, we present novel data also in this aspect from a population sample in an age-group where cardiovascular risks could potentially be reduced and longevity improved. 

If the goal of this investigation is to demonstrate the usability of this technique, the discussion should be more focused on that aspect. Would recommend more attention be given to clinical application of this technique. For example, there are 6 pages of methods on the technique, how translatable are those methods to clinical practice? If the median time to acquire a VCG is 9 minutes (longer than the average face to face time for a patient and a physician in most office visits), how practical is it? Is the additional prognostic information worth the time/effort?

RESPONSE The point is well taken, although the goal was to develop an automatic method to make VCG more usable, not to prove its clinical usability. First, all recordings in this study were performed by staff nurses and there is no need for the presence of a physician. Second, the spatial QRS-T angles are the electrocardiographic parameters with the scientifically strongest documentation of a prognostic value regarding cardiovascular events including all cardiac deaths and specifically sudden cardiac death (ref. #2-11); and the gold standard for their assessment is Frank VCG (ref. # 4). Third, the strongest clinical potential for VCG is presently therefore for risk prediction regarding cardiac death and sudden cardiac death and the identification of candidates for implantable cardioverter defibrillators (ICD) to prevent such events. And the present study is a step in the direction of its clinical application, but we are not yet there. We are therefore a bit hesitant to put too much stress to the potential clinical application but have expanded the text dealing with this issue on lines 516-533.

Reviewer #2: This study aimed to apply a novel method for standardizing the window of analysis for vectorcardiography in a large sample of patients in an epidemiological study in Sweden. Vectorcardiography is based on analysis of 12-lead ECG or Frank VCG recording leads and employs vector-based analyses (i.e., magnitude and angle) of recorded electrical signals. Though not readily employed clinically, the potential of VCG is convincingly great. This specific study in this space was unique, innovative, and largely technically appropriate. The authors demonstrated success of their algorithm in comparison to a randomly selected and manually annotated vectorcardiogram, success in delineating differences between men and women that they saw in their demographic analysis, consistency in serial visits with the same patient, and success in consistently producing a stable signal-averaged QRS complex and T wave for analysis with standard VCG tools. Additionally, the authors recognized the limitations of their study and drew mostly appropriate conclusions from their results.

RESPONSE Thank you. Exactly as you state, the potential of VCG is convincingly great. In order to spread this insight and facilitate the application of VCG, work along several lines is needed. One is spreading the knowledge about what VCG can offer (as expanded upon in the Introduction). Another is the development of methods, which in a standardized and reliable way provide data for different purposes, which is the purpose of this manuscript. Yet another is to present data in a way that can be smoothly integrated in clinical decision making, and we are not yet there. So, this is one step in a direction that seems to be appreciated by the reviewer.

 However, there is a major concern about the motivation and significance of this research. The authors did not fully elucidate the need for such an algorithm nor did they convincingly demonstrate how the application of their algorithm will aid in clinical care. These issues and some technical issues listed below should be considered before acceptance:

RESPONSE In order to clarify the rationale behind this study the second and third paragraphs of the Introduction have been rewritten. Basically, the scientific evidence for the prognostic value of the spatial QRS-T angles is overwhelming, but their clinical application is not established for various reasons of which some are mentioned in the Introduction, neither are any other VCG derived parameters. The rationale for this study is to take one step towards that direction. It should, however, be noted that the development of an automatic method for obtaining VCG derived parameters is only one – although necessary - step in that direction and more is required. As mentioned above there is also need to spread the knowledge about the possibilities with and knowledge about VCG. We have e.g. recently published an article explaining the differences between the two spatial QRS-T angles and between the mean QRS-T angle and the ventricular gradient which both are based on the QRS-area and T-area vectors with the general aim to reduce confusion about some of the central VCG based concepts and measures (ref. #11). It would, however, be presumptuous of us to claim more than that our present study is one central step to provide VCG based measures to use in the future e.g. in the difficult work of reducing the burden of unexpected sudden cardiac death but also in the selection and management of patients with heart failure that are candidates for resynchronization therapy; lines 516-533. 

Major Comments

• Introduction

1. Greater discussion on the application of VCG in clinical scenarios, uniqueness compared to ECG (i.e., what complementary or supplementary information is given by the VCG), and prognostic value need to be established. These are vaguely described in Lines 87-100, but a clear description of VCG, its parameters, and its merits/disadvantages need to be included, especially given its nonstandard use.

RESPONSE The second and third parts of the Introduction has partly been rewritten in response to this comment; please also see the previous response.

A diagram explaining VCG would be helpful, though not necessary as there are various other sources that the authors include for this.

RESPONSE Thanks. We agree that technical details are extremely important and also that there are sources available for detailed descriptions. We therefore prefer to add a recent reference on this topic which includes graphical presentation of electrode positions, P-, QRS- and T-vector loops and calculations from the QRST complexes in the X, Y, Z-leads (ref. #11). And the Academic Editor has actually recommended shortening the text. 

Additionally, the problem with manual selection of VCG segments must be delineated. Without these descriptions of VCG, the significance, novelty, and innovation of the authors’ study is lost.

RESPONSE We are uncertain how to understand this comment. The selection of the most stable segments of the VCG recordings was not manual but automatic. The fixed time-point of the 4th segment for comparison was chosen early in the VCG recording (within 1 min of its start) to test if the extended recording time after the standardized supine rest together with the developed algorithm added to the robustness and reproducibility. The analysis of these segments was thus fully automatic. We have added information on lines 255-260 to clarify this procedure and also in relation to the data presented in Table 3.

• Methods

1. If it is possible to do, the variability between multiple observers (at least two) in manually edited annotation points is an important parameter to include to help demonstrate the success of your automatic algorithm.

RESPONSE The point is taken. For defining the QRST complex, QRSonset and Tend are the critical annotation points of which Tend is well-known to show significant inter-observer variability at manual assessment (Ahnve S: Errors in the Visual Determination of Corrected QT (QTc) Interval…, J Am Coll Cardiol 1985;5:699-702). We have actually performed a meticulous comparison between manual and automatic annotation of T end on VCG recordings before (ref. #17). The text below is part of the Supplemental information provided with that article. In that study the manual annotations from another very experienced electrophysiologist was compared with automatic annotations and the agreement was very good. Table 4 shows that the agreement between the coefficients of variation was very good between the automatic and manual annotation points, as expected from ref. #17. We hope that the reviewer finds this response satisfactory although it is not a comparison between different observers. The results of Dr. Ahnve’s study, which has been confirmed since then, was the reason why only one experienced investigator was used in this study to avoid inter-observer variability in this particular comparison with automatic annotation points as one part of the evaluation of the automatic method. In a recent study by Vink et al. in Circulation 2018;138:2345-2358 (ref. # 26), the authors decided to have one electrophysiologist make the annotation points for the same reason. 

From ref. # 17; Supplement:

“Comparison of manual vs. automatic annotations The computer-set detection points defined according to the algorithms described above were evaluated by comparison with manually set detection points from our database. All annotation points used in this validation process were manually corrected by one person and Tend defined by manually applying the tangent method. In the first comparison 1771 signal-averaged QRST complexes from 1-min samples from 25 healthy subjects studied at supine rest were used. Manual and automatic annotation points for QRS onset and Tend were compared. For QRS onset the mean difference was - 0.16 ms, i.e. the automatic algorithm placed the annotation point for QRS onset slightly earlier than the manual. In 5 QRST complexes (0.3%) the difference was ≥10 ms, and 4 of these deviations were observed in one subject with an indistinct QRS onset. Supplement Fig. 3 shows the comparison for Tend. In 1750 out of 1771 (99%) QRST complexes the Tend-difference was within ±6 ms.

We also compared single (non-averaged) QRST complexes using data from a previous publication on beat-to-beat VR variability in 41 LQTS patients (31 LQT1 and 10 LQT2) in comparison with 41 age- and sex-matched healthy control subjects [3]. Manual and automatic measurements were compared in 5005 single QRST complexes recorded at supine rest. The QT interval could not be defined automatically in 18 out of 1848 available complexes (1.0%) from LQT1 patients and in 18 out of 652 (2.8%) complexes from LQT2 patients. Supplement Table 1 shows a comparison of the QT interval and the Tamplitude as well as short-term-variability (STV) of the QT interval based on measuring QRST complexes during one-minute baseline conditions. While the agreement was excellent in healthy subjects, the differences increased with longer QT intervals and lower Tamplitude. On the other hand, the automatic algorithm differentiated between healthy subjects and LQTS patients more clearly than the manually set QT values.

We conclude that in general the agreement between manual and automatic definition of annotation points and the resulting intervals was excellent.”

2. What is the importance of the instability value and how do you actually calculate it? You discuss that the variability value is calculated from averaging the difference between the fixed and alignment waveforms, and that values of difference greater than 5 uV are used to calculate the instability value, but the actual calculation of instability is unclear. Additionally, the physiologic importance is not well established other than vaguely in Table 5 where “external disturbances” and “internal variations” are mentioned as contributing factors to instability. The process of manually defining something as “external” or “internal” is suspect without a more complete and rigorous analysis. Please describe how you delineated this.

RESPONSE The importance of the instability value is to provide a quantitative measure of the variability of the QRST complexes (cardiac cycles from the electrical perspective) and signal to the responsible investigator that the level of variability warrants manual scrutiny of the selected recording segment and representative 10s-saQRST complex to find out the reason; lines 199-205, 275-278, 389-401, and 476-500. This will always be part of the procedure just as over reading of an ECG with automatic interpretation is necessary before making clinical decisions especially when any pathology is claimed by the ECG system. With our system a warning is issued in the presence of a high instability value. 

Figure 1 illustrates in the upper panel how one QRST complex (blue) is kept fixed while the one that should be compared (red) is superimposed/slided over the fixed complex. The difference in morphology is then showed in the middle panels, and the calculation of the difference in the lower panel with the signal enlarged. 

The flowchart in Fig 2 has been updated to clarify the calculation of the instability value. 

The reason for choosing ≥5μV was related to the resolution (2.5 μV) of the sample size as now explained on lines 201-203. Examples of external (to the study subject) and internal (of physiological or pathophysiological origin within the particular study subject) are now shown in S4 Fig, panels a & b. Please also see the responses below to “Minor comments # Methods 4 & 5.

• Results

1. The significance of a percentage difference in CV in Table 3 is unclear. Though a larger difference in CV would show that there is greater variability in the manual annotation on the fourth saQRST than in your automated algorithm, the significance of a 11.7% vs. 14.7 % difference is unclear. An interpretation or a separate statistic that indicates significant differences would be helpful. Additionally, it is assumed that there will be some inter-beat variability regardless of whether the automated or manual technique is used. Is part of the CV difference due to this inter-beat variability?

RESPONSE Unfortunately, we have not been clear enough and there is a misunderstanding, which has partly been dealt with above in response to the comments starting with “Additionally, the problem with manual selection of VCG segments must be delineated.” 

Table 3 thus shows a comparison of the coefficients of variation of two fully automatically analyzed 10s-saQRST complexes, the best from the entire recording (algorithm selected) versus the 4th or early, which was arbitrarily chosen. So, the main conclusion to be drawn from this comparison is that the reproducibility improves for a majority of parameters by applying the algorithm on the repeated recordings. A non-parametric test (Wilcoxon for matched pairs) showed, however, that the overall difference was not statistically significant (p=0.14). This has been added to the text; lines 372-373. Finally, we agree that part of the CV difference is probably due to the inter-beat variability, which, however, might be completely physiological as discussed elsewhere.

2. As discussed in Methods Comment 2, the reasoning behind classifying an instability value as due to external disturbances or internal sources of variability is essential. Is there a way to demonstrate what each of these would look like in a representative trace? Or are there criteria that your VCG reader used to define this?

RESPONSE The point is well taken and examples are now provided as S4 Fig panels a & b.

• Discussion

1. One of the major questions that remains despite the success of your algorithm/study is the applicability of this technique. First, as you note in the paper, the typical ECG/VCG protocol does not include 5-10 minutes of recording so using this algorithm seems to potentially be impractical/infeasible.

Second, because you chose not to look at how specifically the algorithm did in those with underlying heart disease (especially arrhythmias), the importance and utility of the algorithm is unclear. This is especially true as the VCG is expected to help diagnose transient arrhythmias like AF. As such, the success of the algorithm to identify a stable region of the VCG in these conditions is important, and I am not convinced that the author’s algorithm is capable of accomplishing this. 

RESPONSE As stated in the Introduction (lines106-109) the purpose of this study was not to evaluate the performance of VCG in different clinical conditions, other researchers have done that. On lines 337-342 we, however, discuss that both atrial fibrillation and competing sinus and junctional rhythms affected the instability value because the atrial activity was superimposed, and thereby potentially affected the parameter values. A high instability value may thus be caused by the presence of atrial fibrillation as also brought forward on lines 485-489 in the Discussion. We, however, disagree on the point of using VCG for detecting paroxysmal atrial fibrillation, which is the clinical problem and which we assume the reviewer is alluding to. VCG has not been proven better or worse than any other recording method using at least 3 leads for the similar amount of recording time. There are now event recorders available (including applications in smartphones) that are much better for recording transient arrhythmias such as paroxysmal atrial fibrillation as part of scheduled screening recordings e.g. twice a day with the instruction also to record when palpitations occur. We, however, agree this is a clinically important issue.

Finally, all recordings in this study were performed by staff nurses and there is no need for the presence of a physician. As stated in the Discussion, the time limiting step is the time required to allow for rate adaptation and stabilization of ventricular repolarization. Thus, when any physician wants to get a reliable assessment of any repolarization related parameters e.g. the QT interval a 3-5 min supine and quiet rest is necessary also before recording of the routine ECG, even if that requirement is not too often paid attention to.

Minor Comments

• Abstract

1. QRST complex is not a standard complex according to typical ECG where there is a QRS complex and a T wave. Consider defining this complex in the abstract so the reader understands.

RESPONSE A QRST complex is the full cardiac electrical cycle from the noninvasive perspective and has now in the abstract been defined as QRSonset to Tend to make the meaning clear. The duration of the QRST complex is equivalent to the QT interval but that is only one of the 28 parameters that we derived from the VCG recording.

• Methods

1. Why was the second visit non-standardized? Would this have an effect on secondary results?

RESPONSE The point is taken. All participants had agreed to answer extended questionnaires and to go through several tests at a very tight schedule running over two days (ref. #14). Duplicated tests were optional but chosen to be as convenient as possible to those who volunteered to do that (as stated on lines 138-139). Because we used the coefficients of variation for comparisons between methods for VCG analysis of the same recording the secondary results should not be affected. 

2. Where on the neck and back are electrodes placed? It would be helpful to include anatomical landmarks nearby like the vertebrae, vasculature, etc.

RESPONSE A description using anatomical land marks is actually presented on lines 148-150.

3. What is a signal-averaged QRST complex? Some specifics on this calculation would be helpful as it is hard to understand how 70 seconds could represent 7 10s-saQRST complexes and the order of operations that produces a signal averaged complex is unclear. Additionally, is this averaging the same or different than the process you describe for selecting the representative complex?

RESPONSE A signal averaging process is a standard procedure used to improve the signal-to-noise ratio. Instead of using QRST from a single cardiac cycle or the average of the measures from several cardiac cycles where signal noise might affect the measuring process, we calculated an averaged QRST complex from all cardiac cycles within a 10s segment (as is the standard for routine 12-lead ECG). The number of cycles (Beat count in the tables 3 & 4) vary depending on the heart rate but was on average between 60 and 70 bpm in this cohort. For each minute the number of any 10s-saQRST complex would ideally be 6, unless there were disqualifying arrhythmias. The representative 10s-saQRST complex was thus within a group of 7 similar 10s-saQRST complexes. Then we narrowed in on 50s-segments to work with and this segment contained 5 10s-saQRST complexes from which the most representative was chosen. The reviewer’s assumption last in the comment is correct. 

There is a technique called SAECG (signal averaged ECG) employing the same method but used mainly for detecting late potentials in the QRS complex e.g. in the diagnostic work up for arrhythmogenic cardiomyopathy (previously arrhythmogenic right ventricular dysplasia or cardiomyopathy; ARVC).

4. Can you explain why a sliding absolute difference between a portion of the QRS complex was used rather than a cross-correlation for the alignment steps? 

RESPONSE The sliding absolute difference gives the same weight over the entire part of the cardiac cycle where it is used, here from QRSonset to Tend. This is now explained in the text on lines 195-198. In contrast, cross-correlation gives more weight to differences at high signal levels.

5. Was there a reason that 5 uV was selected as the threshold to add to the instability value?

RESPONSE The reason for choosing ≥5μV was related to the resolution (2.5 μV) of the sample value as now explained on lines187-189 and 198-206.

6. A quick question regarding the annotation. It is clear that the algorithm in this paper is focused on selection of a standard window for analysis of VCG parameters. However, the automatic/manual annotation of these parameters is not clear. Can you describe briefly how these parameters are selected and how sensitive the process is to noise? Perhaps using Supplemental Figure 1.

RESPONSE The point is well taken. The legend to S1Fig now describes how the annotation points were defined automatically. It also includes a description of how the manual scrutiny was performed.

• Results

1. Table 1 is an excellent representation of demographics and shows clear differences in biological sex and various parameters in the study. Additionally, Table 2 excellently portrays the difference in various VCG parameters between men and women. A tertiary analysis of other demographics and VCG parameters seems warranted (especially since the data is available).

RESPONSE We very much appreciate the reviewer’s interest also for how VCG reflects non-invasive electrophysiology and which determinants are important for different VCG derived parameters. We take this as reflecting that the reviewer has accepted our methodology as stated in the very first comment. We agree that both physiological and pathophysiological determinants are relevant and interesting in this context. An expanded analysis of that kind would, however, both require an entirely separate and rather complicated statistical analysis to account for co-variates, and potentially also divert any reader’s attention from the aim and focus of this manuscript, how to proceed to acquire reliable data from a VCG recording. In a recent publication we have e.g. pointed to important determinants for abnormal QRS-T angles apart from sex, i.e. the presence of diabetes, hypertension, and the absence of any known disease and regular pharmacotherapy (ref. #11). – A complete response to this very interesting comment can only be provided with a review article

2. What were the values of the Shapiro Wilk test for normality and how did you decide to accept the null hypothesis (i.e., what value of the Shapiro Wilk test was used for this)? Qualitatively, there appears to be a relatively normal spread so more clarity would be appreciated.

RESPONSE The p-value for the Shapiro-Wilk’s test was <0.05, which has now been added on line 296.

3. Is there a statistic to effectively compare the CV, mean, or s.d. of the two different annotations in Table 4? If so, this information would be useful to the reader to understand the significance of this automated annotation.

RESPONSE We have now applied a robust sign test (Wilcoxon test for matched pairs) to test the differences in coefficients of variation in Tables 3 and 4 and included this in the Statistical Methods (lines 302-303). The result has been included in the appropriate places, lines 372-373 and 380. With this method, there were no significant differences in the comparisons although the p-values differed, 0.14 in Table 3 and 1.0 in Table 4.

6. PLOS authors have the option to publish the peer review history of their article (what does this mean?). If published, this will include your full peer review and any attached files.

Do you want your identity to be public for this peer review? For information about this choice, including consent withdrawal, please see our Privacy Policy.

Reviewer #1: No

Reviewer #2: No

---

## [Decision Letter · Decision Letter 1]

31 Aug 2020

Automatic identification of a stable QRST complex for non-invasive evaluation of human cardiac electrophysiology

PONE-D-20-11644R1

Dear Dr. Bergfeldt,

We’re pleased to inform you that your manuscript has been judged scientifically suitable for publication and will be formally accepted for publication once it meets all outstanding technical requirements.

Kind regards,

Elena G. Tolkacheva, PhD

Academic Editor

PLOS ONE

Additional Editor Comments (optional):

Reviewers' comments:

Reviewer's Responses to Questions

**Comments to the Author**

1. If the authors have adequately addressed your comments raised in a previous round of review and you feel that this manuscript is now acceptable for publication, you may indicate that here to bypass the “Comments to the Author” section, enter your conflict of interest statement in the “Confidential to Editor” section, and submit your "Accept" recommendation.

Reviewer #1: All comments have been addressed

Reviewer #2: All comments have been addressed

2. Is the manuscript technically sound, and do the data support the conclusions?

Reviewer #1: Yes

Reviewer #2: Yes

3. Has the statistical analysis been performed appropriately and rigorously? 

Reviewer #1: I Don't Know

Reviewer #2: Yes

4. Have the authors made all data underlying the findings in their manuscript fully available?

Reviewer #1: Yes

Reviewer #2: Yes

5. Is the manuscript presented in an intelligible fashion and written in standard English?

Reviewer #1: Yes

Reviewer #2: Yes

6. Review Comments to the Author

Reviewer #1: (No Response)

Reviewer #2: All comments have been adequately addressed, and there are no new concerns. The manuscript should be accepted for publication.

7. PLOS authors have the option to publish the peer review history of their article (what does this mean?). If published, this will include your full peer review and any attached files.

Reviewer #1: No

Reviewer #2: No

---

## [Editor Report · Acceptance letter]

4 Sep 2020

PONE-D-20-11644R1 

Automatic identification of a stable QRST complex for non-invasive evaluation of human cardiac electrophysiology 

Dear Dr. Bergfeldt:

I'm pleased to inform you that your manuscript has been deemed suitable for publication in PLOS ONE. Congratulations! Your manuscript is now with our production department. 

Kind regards, 

on behalf of

Dr. Elena G. Tolkacheva 

Academic Editor

PLOS ONE